# SpPdp11 Administration in Diet Modified the Transcriptomic Response and Its Microbiota Associated in Mechanically Induced Wound *Sparus aurata* Skin

**DOI:** 10.3390/ani13020193

**Published:** 2023-01-04

**Authors:** Isabel M. Cerezo, Olivia Pérez-Gómez, Rocio Bautista, Pedro Seoane, M. Ángeles Esteban, M. Carmen Balebona, Miguel A. Moriñigo, Silvana T. Tapia-Paniagua

**Affiliations:** 1Instituto de Biotecnología y Desarrollo Azul (IBYDA), Faculty of Sciences, Department of Microbiology, University of Malaga, 29010 Málaga, Spain; 2SCBI, Bioinformatic Unit, University of Malaga, 29590 Málaga, Spain; 3Faculty of Sciences, Department of Molecular Biology and Biochemistry, University of Malaga, 29010 Málaga, Spain; 4CIBER de Enfermedades Raras (CIBERER), 28029 Barakaldo, Spain; 5Campus Regional de Excelencia Internacional “Campus Mare Nostrum”, Faculty of Biology, Department of Cell Biology and Histology, University of Murcia, 30100 Murcia, Spain

**Keywords:** aquaculture, fish, gilthead seabream, microbiota, probiotic, RNA seq, skin, wound

## Abstract

**Simple Summary:**

This study evaluated using NGS techniques and bioinformatic analysis the influence on the microbiota and transcriptomic response of skin mechanically wounded gilthead seabream (*S. aurata*) specimens fed with a probiotic SpPdp11 supplemented diet. Four group of fish were established: non-wounded and wounded fed control diet, and non-wounded and wounded fed a probiotic enriched diet. The wounded group that received the probiotic diet showed a decrease in the abundance of taxa related to bacterial biofilm formation, and transcriptomic results suggested that specimens of this same group had a group of genes up-regulated and down-regulated related differently to those expressed in control group (non-wounded). Then, five genera that presented significant differences between these groups showed positive correlations with genes related to cell migration and negative correlations with inflammation and cell proliferation. These results are promising, and they open new perspectives and possibilities in the use of probiotic SpPdp11 to improve the skin after an injury, which can happen frequently in farmed specimens.

**Abstract:**

Skin lesions are a frequent fact associated with intensive conditions affecting farmed fish. Knowing that the use of probiotics can improve fish skin health, SpPdp11 dietary administration has demonstrated beneficial effects for farmed fish, so its potential on the skin needs to be studied more deeply. The wounded specimens that received the diet with SpPdp11 showed a decrease in the abundance of *Enterobacteriaceae*, *Photobacterium* and *Achromobacter* related to bacterial biofilm formation, as well as the overexpression of genes involved in signaling mechanisms (*itpr3*), cell migration and differentiation (*panxa*, *ttbk1a*, *smpd3*, *vamp5*); and repression of genes related to cell proliferation (*vstm4a*, *areg*), consistent with a more efficient skin healing processes than that observed in the wounded control group. In addition, among the groups of damaged skin with different diets, *Achromobacter*, f_*Ruminococcaceae*, p_*Bacteroidetes*, *Fluviicola* and *Flavobacterium* genera with significant differences showed positive correlations with genes related to cell migration and negative correlations with inflammation and cell proliferation and may be the target of future studies.

## 1. Introduction

The quick development of aquaculture due to the high demand for fish had led to intensive farming and conditions that have a significant impact on fish’s health [1]. In this context, the mucosal surfaces of fish play a key role because they are the barriers where interactions between the host and the aquatic environment take place. The microbiota housed on these surfaces perform functions relevant to fish health. The epidermis of fish is involved in the homeostasis of the organism and consists of several layers of living cells with keratinocytes being the most abundant, and among them are other cell types including goblet, sensory and alarm cells. The outer (apical) part of this peculiar epithelium is covered by a layer of mucus (secreted mainly by goblet cells) and an ex-tense microbiota above it [2]. In addition, mucus and epithelium contain many antimicrobial factors, such as proteases, complement, lysozyme, C-reactive protein, antibodies, transferrin, lectins, polypeptides and antibiotics [3,4,5]. In addition, the skin microbiota is a complex and diverse set of microbial communities that interacts closely with the surrounding environment and has fundamental activities in the development and maintenance of fish physiology and health [6] defending against pathogens [7,8] and by its interaction with the immune response [9,10]. Skin microbiota is more exposed to the circulating environment. Factors such as diet and stress can induce changes in the composition [11,12] and in mucosal skin secretions [13]. So it is more susceptible to alterations than the gut microbiota [14,15]. Understanding the skin microbiome would provide a better understanding of host–microbe interactions, which would have important implications for prophylactic measures in aquaculture.

Skin damage is a relatively frequent symptom associated with intensive fish farming and it is can be associated with a higher susceptibility of the host to disease outbreaks [16,17], developing skin wounds and/or abrasions, or inflammation of epithelial cells [18]. Damage to the skin of fish has an impact on the economic value of species due to their reduction of biological productivity and appearance which causes consumers to avoid buying the product. In injury sites of diseased fish, the mucosal barrier is expected to be compromised, with a substantial presence of potential pathogens. For these reasons, there is an increased interest in fish skin health in recent years as it provides mechanical protection against physical, chemical and biological damage. In this sense, mucus also plays a relevant ecological role as fish shoaling or alarm signals and in interspecific interactions, such as prey–predator relationships, parasite–host interactions, and symbiosis [19,20]. In this context, the analysis of the skin microbiota of fish for the identification of strategies to enhance the prevention and treatments of skin wounds and seek potential microbial biomarkers of fish skin health [21].

Many scientific approaches are aimed at understanding skin epithelium processes and how to reinforce its integrity, as well as the rest of its components [22]. One of these approaches is the use of probiotics, which can enhance the health of the fish skin [23,24]. Several probiotic mechanisms are known to exert beneficial effects on hosts at the gut level, but their role on the skin is still unknown [25,26]. *Shewanella putrefaciens* Pdp11 (SpPdp11) is a probiotic bacteria isolated from healthy skin of *S. aurata* [27], whose dietary administration had beneficial effects on farmed fish, such as *S. aurata* and *Solea senegalensis* [28]; studies have addressed the impact of SpPdp11 in the intestinal microbiota of aquaculture species [29], observing the ability of the probiotic SpPdp11 to reduce the presence of species described as potential pathogens [30,31]. However, the potential effect on skin microbiota after ingestion of this diet has not been studied yet. Some progress as described by Chen et al., 2020 [23] who reported the effect of the dietary administration of SpPdp11 on the transcription of some genes related to the immune system’s regulation and tissue repair on wounded skin of *S. aurata*. Furthermore, the potential changes induced in the wounded skin microbiota by SpPdp11 and its effect on host’s skin general transcriptomic response were not analyzed. RNA sequencing (RNA-seq) of healthy and mechanically injured skin and its associated microbiota was performed using NGS technologies to provide new insights into the effect of probiotic administration and possible processes involved in skin repair in their hosts from these same fish.

Therefore, the aim of this study is to observe the effect of a diet supplemented with the probiotic SpPdp11 on the skin mucus-associated microbiota and transcriptomic response in mechanically wounded *S. aurata* specimens.

## 2. Material and Methods

### 2.1. Bacterial Growth

The SpPdp11 strain was grown in tryptone soya broth (OXOID Ltd., Basingstoke, UK) supplemented with 1.5% NaCl (TSBs, 20 h, 20 °C) under continuous shaking. Contant shaking was used to wash the bacteria that were removed from the plates in sterile phosphate-buffered saline (PBS, pH 7.4). The density of the bacterial suspension was resolved utilizing Coulter Z2 particle counter (BECKMAN Coulter, Barcelona, Spain), and the volume was adjusted to the target concentration (10^9^ cfu g^−1^). This dose was chosen based on the health benefits in *S. aurata* and *S. senegalensis* as reported in previous studies [27,29]

### 2.2. Animals, Experimental Design, and Sampling

A total of 48 specimens (21.8 ± 0.9 g mean body weight) of the hermaphroditic protandrous seawater teleost gilthead seabream (*S. aurata L*.) were obtained from a local farm (Murcia, Spain) and kept in quarantine for four weeks and were fed a commercial pellet diet (Skretting) at 2% rate of fish biomass per day. Afterwards, the fish were randomly assigned to four running seawater aquaria in the Marine Fish Facilities at the University of Murcia (250 L, with continuous aeration and flow rate 900 L h^−1^). Two tanks were fed twice a day (7 a.m. and 7 p.m.) with control diet and the other two with probiotic diet at the previous indicated rate. The fish were maintained with an artificial 12 h light/12 h dark photoperiod and water parameters of 28% salinity and 20 °C. The basal feed was the commercial diet, that was supplemented with equal volumes of phosphate saline buffer (PBS) or bacterial suspension (10^9^ cfu SpPdp11 g^−1^) to create the control and probiotic diets, respectively.

As shown in Figure 1, after 30 days of assay, eight fish from each diet (four from each aquarium) were sampled (C and P groups) by taking skin samples from the middle of the left flank above the lateral line (with an 8 mm diameter biopsy punch and 2 mm deep). Before sampling, the fish were euthanized with 100 mg L^−1^ of clove oil (Guinama^®^). The remaining fish (eight specimens from each diet, four from each aquarium) were anesthetized (20 mg L^−1^ of clove oil) and similar 8 mm diameter and 2 mm deep wounds were made in the middle of the left flank (below the lateral line) on the skin of all of them, with a circular biopsy punch (Integra™ Miltex™), taking care in avoiding any possible contamination with urinogenital and/or intestinal excretions. This animals remined in their respective aquaria receiving their corresponding diet for an extra seven days. The fish were fed the initially assigned diet, being the daily rate adjusted accordingly to the fish weight. Seven days after the wounding, eight specimens from each diet (four by aquarium) were also sampled (CW and PW groups). Skin samples were obtained from the same body part of uninjured fish. Skin samples of four fish per experimental group were randomly selected and each sample was divided into two for subsequent RNA and DNA analysis and kept at −80 °C until use.

### 2.3. DNA Extraction and Sequencing of the 16S rRNA Gene from the Skin-Associated Microbiota

The skin samples (*n* = 4 per group), stored at −80 °C were thawed gradually on ice and the mucus contents were extracted by pressing towards the ends with a sterile object. After homogenizing, a portion of 50 mg of each mucus sample was used for DNA extraction. DNA was extracted in all cases following the protocol based on saline precipitation by [32], with minor modifications [31]. DNA was quantified fluorometrically with Qubit™ dsDNA HS Assay Kit (Thermo Fisher Scientific, Waltham, MA, USA) and by spectrophotometric and electrophoretic methods to study the degree of purity, quality and integrity of the DNA.

16S rRNA of mucus samples was sequenced on Illumina MiSeq platform (Illumina, San Diego, CA, USA) with 2 × 250 bp paired-end sequencing in the Ultrasequencing Service of Novogene Europe (Cambridge, United Kingdom) following the 16S Metagenomic Sequencing Library Preparation protocol. Sequencing was carried out using the sense primers 341F and 806R (5′ CCTAYGGGRBGCASCAG 3′ and 5′ GGACTACNNGGGTATCTAAT 3′ respectively) [33,34], directed to the variable regions V3-V4 of the 16S rRNA gene. All Illumina reads were analyzed with FastaQC software (version 0.11.4) and Q30 score was <92%. Further data processing including trimming and 16S analysis and visualization was performed with the workflow based on the MOTHUR software package (version 1.39.5). Paired-end reads of each sample were processed into Mothur (version 1.39.5) to remove low-quality reads and homopolymers strings. The chimeras were detected and removed using UCHIME (version 4.2.) [35]; following, the remaining representative, non-chimeric sequences were aligned and clustered into operational taxonomic units (OTUs) against the Greengenes database (version 13.5) with 97% identity cutoff, the total count threshold was set at 0.005%. Abundance of operational taxonomic units (OTUs) of the skin microbiota was processing using *phyloseq* [36] and *vegan* library [37] in R statistical package. Readings were normalized based on rarefaction curves and singleton were removed. In addition, it was calculated the coverage using the Good’s coverage coefficient, as well as the ecological indexes. Alpha diversity is estimated using the Chao1 and the Shannon and Simpson indexes, to assess richness, diversity and dominance, respectively. For statistical analyses between diversity indexes, two-way ANOVA (*p* < 0.05) was identified. Differential abundance of taxa was carried out using the R package DESeq2 with a false discovery rate (FDR < 0.05).

### 2.4. RNA Isolation, Quality Control and Sequencing

Total RNA was extracted from gilthead seabream skin samples using TRIsure™ (Qiagen, Germany) methodology and eluted in nuclease-free water Total RNA concentration was quantified using a Qubit™ RNA Broad Range Assay Kit (Thermo Scientific, USA) (1742.09 ± 628.12 ng/µL) and stored at −80 °C to posterior analysis. Prior to RNA sequencing, RNA integrity and quantitation were assessed using RNA Nano 6000 Assay Kit of the Bioanalyzer 2100 system (Agilent Technologies, Santa Clara, CA, USA). RNA sequencing libraries were performed by Sequencing Center Novogene (Cambridge, United Kingdom). RNA total library is prepared for UltraTM RNA Library Prep Kit for Illumina^®^ (NEB, Ipswich, MA, USA) starting from 1 µg of RNA per sample. The twelve libraries were sequenced using Hiseq2500 Illumina instrument (paired-end length sequencing 150). The raw reads were processed in removing sequencing adapters, poly-N regions and low-quality reads by CASSAVA software. In addition, the Q20, Q30 and GC content of the clean data was calculated. All subsequent analyses were performed based on the clean, high-quality data. Only three of the samples passed the sequencing quality control, so an n=3 per group was kept for RNA seq analysis.

### 2.5. Differential Expression Analysis and Functional Enrichment

Paired-end clean reads were mapped to the *S. aurata* reference genome (fSpaAur1.1, https://www.ncbi.nlm.nih.gov/assembly/GCF_900880675.1/ (accessed on 30 July 2019)) using HISAT2 software [38]. Transcripts counts were obtained using HTSeq software [39]. The level expression of each gene (counts) was expressed as the number of Fragments Per Kilobase Million (FPKM). Differential expression studies were carried out using DESeq2 Package R [40]. Candidate to differential expressed transcripts (DEGs) were those with an absolute log2 fold-change (logFC) value ≥ 1.3 and false discovery rate (FDR) < 0.05, in all study conditions. DEGs of the comparison PW vs. CW were annotated with Ensembl orthologous genes in *D. rerio* (*Danio rerio*, Ensembl genome browser 105) using Full-LengtherNEXT [41]. These orthologues were used to carry on the enrichment analysis using ExpHunter Suite [42]. The Gene Ontology (GO), KEGG pathway, and Reactome terms kept as significant categories were those with a *p* adjusted value ≤ 0.05.

### 2.6. Microbiota and Genes Correlation

Previously, three samples of microbiota and RNA-seq were selected from each individual. A correlation between OTUs abundances and gene expression level was calculated for the PW vs. CW comparison. For this, 16S marker gene reads filtered by rarefaction curves and genes expression level matrix (FPKM) were normalized by DESeq2′s median of ratios. The Spearman rank correlation [43] was calculated using the *rcorr* function of the R *Hmisc* package [44]. Significant OTU-gene correlations were considered at *p* < 0.05, and only those correlations were selected with a value of |r| > 0.9. Cytoscape v3.8.2 was used to visualize the significant correlations present in the top 20 genes with the highest rate of change and OTUs with relative abundance > 0.05%.

### 2.7. Ethics Approval and Consent to Participate

All fish-related studies were carried out in strict accordance with European Union (2010/63/UE) and Spanish (RD 1201/2005 and RD 53/2013) guidelines for the use of laboratory animals. The experiment with fish were developed at the University of Murcia (Spain) and were authorized by its Ethical Committee (protocol code A13150104). The procedures were also approved by the Bioethics and Animal Welfare Committee of the Institute of Agricultural and Fisheries Research and Training (IFAPA) and given the registration number 17/11/2016/171 according to the national authorities for regulation of animal care and experimentation.

## 3. Results

### 3.1. Skin Microbiota Analysis

Four replicates were analyzed per group. The sequencer generated a total of 2,166,953 raw. These reads were processed bioinformatically, generating a total of 111,408 valid reads, being 87,843.44 sample mean, that clustered in 1332 OTUs. The results of alpha diversity were shown at the level of richness (Chao1), diversity (Shannon) and dominance (Simpson) index, and non-significant differences were observed among the different experimental groups (Two-way ANOVA, *p* > 0.05) (Table 1, see more details in Appendix A).

The reads were filtered using the rarefaction curves to the minimum value of valid asymptotic readings (42,673 reads, 1244 OTUs) (Appendix A). Library coverage was calculated using Good’s coverage with a result of 99 ± 0.02 % (SEM ± standard deviation). To eliminate clustering errors, the samples were filtered by removing singleton and doubleton, a total of 680 OTUs were analyzed. Sequences related to *Proteobacteria, Actinobacteria*, and *Bacteroidetes* phyla were the most abundant, representing approximately 70% of the total sequences analyzed in all cases (Figure 2 and Appendix A), while *Actinobacteria, Acidimicrobia*, α-, β-, and γ-Proteobacteria were the most frequent classes in all experimental assays (Figure 3 and Appendix A). Notably, a comparative analysis at genera level (abundance > 0.5%) showed a high value of unidentified genera (Figure 4 and Appendix A).

Differences were not detected between fish (DESeq2, *p* < 0.05) from C and P groups or C and CW groups, whereas they were found between P and PW (Appendix A) and CW and PW groups (Figure 5). The significant differences in abundances of OTUs, such as *c_Clostridia* unclassified, *Ruminococcus*, f_*Ruminococcaceae*, *Propionibacterium*, *Idiomarina*, *Photobacterium*, *Achromobacter*, f_*Enterobacteriaceae*, *Fluviicola* and p_*Bacteroidetes* unclassified were observed when skin microbiotas of fish from P and PW groups were compared. In CW and PW comparison, the results also showed a significantly lower abundance of OTUs associated to phyla as Firmicutes (c_Clostridia unclassified, *Ruminococcus* and f_*Ruminococcaceae*), Actinobacteria (*Propionibacterium*) and mainly Proteobacterias (*Salinisphaera, Alcanivorax, Idiomarina*, *Chromohalobacter*, *Photobacterium*, *Achromobacter* and f_*Enterobacteriaceae)*. On the other hand, fish of the PW group showed a significantly higher abundance of OTUs related to Bacteroidetes phylum, such as *Sediminibacterium*, *Flavobacterium*, *Fluviicola* and p_*Bacteroidetes unclassified*.

### 3.2. Skin Transcriptome Sequencing and Analysis of Differentially Expressed Genes (DEGs)

A total of 310,183,755 raw reads were recovered with Illumina sequencing of three replicates per group. The final number of reads per individual ranged 50,971,570.50 ± 5,570,365.99 (means ± deviation), with a total of 81.67 ± 23.28 % reads mapped. The number of reads in each treatment group was well balanced with over 40 million in each group (Appendix A). FPKM whose with a value greater than 0.3 were taken as valid transcripts, removing the lower readings. Over 16200 genes were expressed in which approximately 400 genes exceeded the value of 100 FPKM (Figure 6A). The study of FPKM > 0.3 per sample revealed a clear clustering of the gene expression profile in the case of samples from CW group (Figure 6B), while a grouping was observed between most of the samples from C and PW. On the other hand, samples corresponding to P group were also clearly clustered (Figure 6B). To identify genes with significantly modified transcription (DEGs) the RNA-seq profile obtained from fish of C group was compared with specimens of P and CW groups. Analysis of DEGs revealed that the experimental wounding in the skin produced the down-regulation of 693 genes and up-regulation of 952 genes (DESeq2, *p* adjust < 0.05), while the inclusion in the diet of SpPdp11 (P treatment) caused, in the skin, the down- or up-regulation of 35 and 137 genes, respectively, (DESeq2, *p* adjust < 0.05, Appendix A). CW and PW groups were also confronted; the comparison between these groups revealed that 44 genes were differentially overexpressed while 72 were down-regulated (DESeq2, *p* adjust < 0.05, Figure 6C).

### 3.3. Analysis of DEGs with the Highest Expression Change Value

Wounding did not show significant changes at the skin mucus-associated microbiota with fish feed with control diet. The group that produced the most significant changes was the PW, with a different microbial profile to the other study groups. To compare how diet acted against wounding, the CW and PW comparison was used for the rest of the analyses. The transcription of top 10 up-regulated and top 10 down-regulated genes in specimens of PW group in comparison with CW group was studied and the results are summarized in Table 2. It was possible to observe that in PW there was an up-regulation of genes involved in: (i) acylation and methylation of histones and gene activation, such as *brd2a* and *kmt2cb*; (ii) encoding information for proteins implied in cell signaling (*adgrl1a*, *itpr3* and *adamtsl5*) and cell migration as *smpd3*, *vamp5*, *pxna* and *ttbk1a*; (iii) regulation of the innate immune system (like-*nlrc3)*. On the other hand, it is observed that in these same fish was detected the down-regulation of genes encoding information for apoliproteins (*apoa4b*.2, *apoeb* and *apol*), enzymes involved in inflammation and tissue repair (*chia*.1 and *chia*.3), cell proliferation (*areg, meltf* and *vstm4a*), regulation of mucus (*gcnt3*) and proteins implied in anchoring in extracellular matrix (*vwa2*) (Table 2).

### 3.4. Enrichment Analysis of S. aurata Transcriptome and Its Differentially Expressed Genes

Based on the objective of this work, the DEGs between the damaged groups CW and PW was studied in detail using DEgenes hunter workflow. From the 12.288 sequences of the *S.aurata* transcriptome, 83.01% were related with a *D. rerio* orthologue. The 116 DEGs obtained from the comparison between fish from CW and PW groups were analyzed with GO, KEGG and Reactome databases. Different GO categories were enriched. “Cell substrate binding” represented a significant functional enrichment in “Biological Processes” (*p* adjust *<* 0.05, Appendix A). Genes related with calcium channel activities and motor proteins, such as myosin were overexpressed, although only extracellular matrix binding function was significant in the “Molecular function” category (*p* adjust *<* 0.05, Figure 7). In addition, in the “Cellular Component” category were repressed in relation to the extracellular matrix containing collagen, and the polymer cytoskeleton fiber (*p* adjust *<* 0.05, Appendix A). Regarding the KEGG categories, the enriched functional analysis did not show significant categories (*p* adjust > 0.05) (Appendix A) while in Reactome analysis the insulin-like growth factor-binding protein (IGFBP) genes have a significant enriched *p* adjust < 0.05 (Appendix A).

### 3.5. Correlation between the Skin Mucus-Associated Microbiota and the Most Up- or Down-Regulated Genes in Wounded S. aurata

A total of 116 DEGs and 680 genera showed a total 733 significant correlations (Spearman, *p* < 0.05 and |r| < 0.9, Appendix A), but a study in detail was carried out correlating the 20 DEGs showing the highest up- and down-regulations in PW group in comparison with CW group and the OTUs obtained in the microbiota comparative analysis of skin of PW and CW groups (abundances < 0.5%). In Figure 8 are represented the 11 OTUs and 11 DEGs showing 24 significant correlations (11 positive and 13 negative). *ttbk1* and like-*nlrc3* genes were those showing the highest number of correlations with different OTUs, specifically 5 (all negative) in the case of *ttbk1* and 4 (3 negative and 1 positive) from like-*nlrc3*. A total of 5 out of 11 OTUs (*Achromobacter, f_Ruminococcaceae, p_Bacteroidetes, Fluviicola* and *Flavobacterium*) showed significantly different abundances between specimens from CW and PW groups, while *Achromobacter* and *f_Ruminococcaceae* showed significantly decreased abundances in PW specimens, the abundances of *p_Bacteroidetes, Fluviicola* and *Flavobacterium* were significantly increased in specimens of this same group. These OTUs included 45.83% out of 24 correlations stablished, specifically 6 positive (with *chia.1, gcnt3, vstm4a, vamp5, itpr3* and *adamtsl5* genes) and 5 negative (two correlation with *ttbk1* and *chia.3* and one with like-*nlrc3* genes) correlations.

## 4. Discussion

Analysis of the skin mucus-associated microbiota of the different groups of fish showed that dietary administration of SpPdp11 and mechanical wounding did not induce significant changes in alpha diversity (Two-Way ANOVA, *p* < 0.05) regardless of the treatment analyzed. In contrast with these result where the mucus skin was not directly of the wound, others studies the diversity of the microbiota associated with the fish skin wounded by microbial infection was increased [1,21]. It possible that the realization of a mechanical wound does not imply changes in microbial diversity indirectly, that does happen when ulcers are produced by biological agents and its interactions with the microbiota.

The taxonomical composition to *phylum* level was very similar in all treatments applied, being Proteobacteria, Actinobacteria and Bacteroidetes the most abundant groups. These results agree with previous studies carried out to analyze the skin microbiota of *S. aurata* even in those specimens with artificially damaged skin [45]. The mechanically induced wound or the dietary administration of SpPdp11 did not induced significant changes on the skin mucus-associated microbiota of *S. aurata* specimens. On the contrary, when wound was induced to fish receiving the probiotic diet (PW) a significant modulation of their skin microbiota was observed in comparison with specimens of P and CW groups. PW showed a decreasing OTUs abundances, such as Clostridia class, and unclassified *Ruminococcaceae* and *Enterobacteriaceae* families, and *Idiomarina*, *Photobacterium*, *Achromobacter* and *Ruminococcus* genera. In humans, some of the mentioned genera are facultative anaerobes, especially these microorganisms have been reported to be significantly associated with wounds and their presence is considered as a pessimistic prognosis regarding healing in the wound microbiome [46]. On the other hand, PW fish showed significant decreases of *Propionibacterium*. This genus is known by its capability to produce propionic acid [47], which is associated with delay in the wounded healing [48,49]. In addition, the presence of taxa related to *Enterobacteriaceae* [50], *Photobacterium* [51,52] and *Achromobacter* [53] are related to form bacterial biofilm. In previous studies, biofilm has an important factor affecting healing by inhibition of proliferation and migration of epithelial cells [54]. This fact has not been studied in *Sparus aurata* but in the future, some studies could be focused to corroborate the relation between the presence of those taxa and epithelial cell proliferation. This decrease could be due to the high level of proteases on wounded *S. aurata* after being fed with SpPdp11 supplemented diet, as previously described [23]. Authors discussed proteases might affect bacteria involved in skin damage production, cleaving their proteins, interfering in their adhesive capability, and affecting their survival [2].

Wound repair is a complex biological phenomenon, dependent upon protein synthesis, matrix deposition, cellular migration, and replication [55]. These processes are guided by signals of extracellular matrix molecules and peptide growth factors. In this context, the comparison between CW and PW showed a decrease of DEGs related to the collagen matrix and cytoskeleton in fish of PW group, that are essential in wound repair [56,57,58]. On the contrary, specimens of the PW group showed up-regulation of genes involved in the calcium metabolism, such as *itpr3* encoding receptors for inositol 1,4,5-trisphosphate that mediates intracellular calcium release with a relevant effect in the normal homeostasis of skin [59,60,61]. In addition, these same fish of PW group showed the up-regulation of genes, such as *panxa*, *ttbk1a*, *smpd3* and *vamp5* involved in cell migration [62,63], in protein localization [64] and in the regulation of cell proliferation [65]. Similarity, *adgrl1a* and *adamtsl5* genes, involved in cell interaction and signaling were also up-regulated in PW fish [66,67].

On the other hand, genes related to the epigenetic effect, such as the histone acetylation (*brd2a*) [68,69] and methylation (*kmt2cb*) [70] were also up-regulated in PW specimens. It could have a key effect in epidermal stratification, proliferation, and differentiation modulating keratinocytes during skin healing. In addition, the protein encoded by *brd2* acts as a transcriptional activator by the acetylation marks on histones by bromodomain (BRD) proteins with strong anti-inflammatory properties [71,72]. However, PW showed a reduction of DEGs related to apoliproteins (*apoeb*, *apoa4b.2* and *apol*), chitinases (*chia.3*, *chia.1*) involved in immunological response, wound healing by interfering with inflammation and tissue repair [73,74] and other genes, such as *meltf*, *areg*, *gcnt3*, *vwa2*, and *vstm4a*. Some of these genes have been linked to intestinal tissue regeneration [75], early healing [76] and cell proliferation [77,78,79]. Chen et al. (2020) reported a significant down-regulation of the transcription of genes encoding proinflammatory cytokines (il-1β, il-8) and anti-inflammatory cytokine (tgf-β) in the skin of wounded *S. aurata* specimens fed a diet supplemented with SpPdp11 in agreement with our study, whose chitinases have been repressed in the fish of PW group.

The correlations can help to distinguish possible relations between bacterial taxa and changes in number of host cell transcripts. This does not imply causality but is a good starting point and, in the future, the effect of manipulating specific bacterial taxa on the expression of these genes could be studied. In this context, when correlation between microbiota and the top 20 genes was stablished, it was possible to determine that 11 OTUs were associated to positive or negative correlations with the expression of these genes, and 3 of them, *Achromobacter, Fluviicola, f__Ruminocacceae* and *p_Bacteriodetes*, showing significant differences of their abundances when microbiota of PW group was compared with microbiotas from CW and P groups. *ttbk1* and *nlrc3* were the gene with the highest number of interactions. The transcription of *nlrc3* that encodes a protein that inhibits the activation of NF-κβ factor, playing an inhibitory role during the inflammation [80] and blocking the cell proliferation [81], was positively correlated with DA10 genus, while it was negatively correlated with *Methylobacterium, f__Ruminocacceae* and *Paracoccus*. As previously explained, this gene encodes a protein that plays an inhibitory role during the inflammation and blocks the cell proliferation. Related to these results, it has been reported that *Paracoccus* was positively correlated to the systemic transcription of proinflammatory cytokines, such as Il-8 and TNFα, in specimens fed diets containing novel protein sources [82]. These results agree with the lower abundances of this genus detected in the skin microbiota of fish from the PW group compared to those detected in those from the CW group. OTUs related to *Methylobacterium, f__Ruminocacceae* and *Paracoccus* besides to *f__Halomonadaceae* and *Achromobacter*, with abundances significantly different between fish of CW vs. PW groups, were negatively correlated with the transcription of *ttbk1* involved in cell migration [83]. *Achromobacter* forms biofilm an important factor by to inhibit the migration of epithelial cells [54]. The significant reduction of the abundance of this genus in the skin microbiota of fish from PW could be related with the higher transcription of this gene in these fish. In addition, *Achromobacter* produces metalloproteases, such as collagenases [84,85] which can inhibit the cell migration [86].

The presence of *f__Sphingomonadaceae* unclassified showed a positive correlation with *areg* and *apoa4b*, genes involved in early healing [76]. In other study, *Sphingomonas* was informed as one of the proinflammatory genus whose abundance was reduced in a murine wound healing model treated with collagen peptides isolated from fish skin [87]. *Fluviicola* (a Bacteroidetes genus) and Bacteroidetes unclassified *phylum*, with abundances increased in PW group vs. CW, exhibited negative and positive correlation with *chia-1* and *vamp5*, respectively, whereas *Achromobacter* showed a positive correlation with the transcription of *chia-1* and *vstm4* and *gcnt3*, gene involved in cell proliferation and inflammation and tissue reparation. Bousbaine et al. (2022) [88] described a β-hexosaminidase, a conserved enzyme of the Bacteroidetes *phylum*, that in a mouse model of colitis, protected against intestinal inflammation, and it could be related to the negative correlation observed with regard to the transcription of *chia-1* gene. On the contrary, *Achromobacter* species have been associated with inflammation [89], and it could be related with the production of biofilm by this genus, ability related with the possibility to induce inflammation [90].

On the other hand, *Flavobacterium* showed positive correlation with the transcription of *adamts15* and *itpr3*, genes encoding proteins involved in cell signaling. There are microorganisms that can stimulate cell calcium flux in response to messengers by ITRP3 through bacterial pore-forming proteins (PFPs) [91]. Some species of *Flavobacterium* have been reported as producers of PFPs [92] and the increases of abundances of *Flavobacterium* in fish from PW group could be involved in this stimulation of the calcium flux in skin cells.

Naya-Català et al., (2021) [82] reported a systemic effect of intestinal microbiota in specimens of *S. aurata* fed different diets based on correlations of the transcriptomic of liver and head kidney with changes in the intestinal microbiota. SpPdp11 has demonstrated capability to modulate gut microbiota of *S. aurata*. In this study, it has been possible to stablish that the dietary inclusion of the probiotic SpPdp11 had a modulator effect on the microbiota and transcriptomic response of wounded skin of *S. aurata* and it could suggest the existence of a gut–skin axis as well as it has been described in mammals [93,94]. However, more research is necessary to demonstrate this suggestion, such as increasing the number of replicates in the study.

## 5. Conclusions

In summary, analysis of the skin mucus-associated microbiota in mechanically wounded *S. aurata* specimens reveled there was no effect of fish feed on the wound with the control diet. On the contrary, the probiotic SpPdp11 diet showed a modulation on wounded fish (PW), especially by increases of abundances of OTUs related to Bacteroidetes and decreases of abundances of OTUs related to *Achromobacter, Paracoccus, Photobacterium, Propionibacterium, Idiomarina* and *Ruminococcus*, among others. On the other hand, differences in skin transcriptomic of genes involved especially in cell migration, cell proliferation and inflammation were also observed between CW vs. PW groups. Finally, several genera that showed significant differences in their abundances in PW showed positive correlations with genes related to cell migration and negative correlations with inflammation and cell proliferation.

## Figures and Tables

**Figure 1 animals-13-00193-f001:**
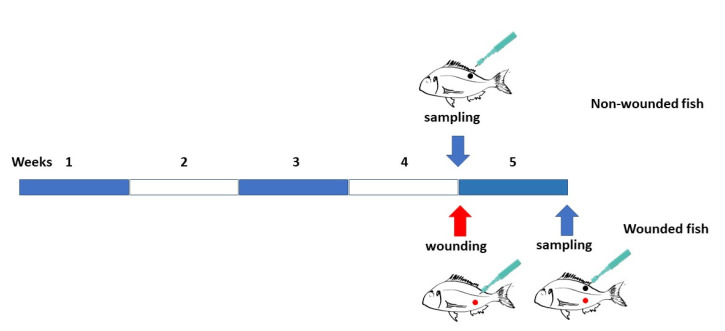
Experimental design. Gilthead seabream were fed for 30 days with a control diet or enriched with SpPdp11 10^9^ cfu per g of diet. At 30 days of assay, after sedation, 8 fish per diet (non-wounded) were sampled by taking skin samples from the middle of the left flank above the lateral line (with an 8 mm diameter biopsy punch and 2 mm deep) obtaining samples from groups C and P. The rest of fish on the left flank, similar wounds were made with the same biopsy punch but below the lateral line and fish were fed for an extra week with assigned diet. After 7 days, 8 fish per diet were anesthetized and skin samples were taken from the same location described and with the same procedure as described in the non-wounded fish (groups CW and PW).

**Figure 2 animals-13-00193-f002:**
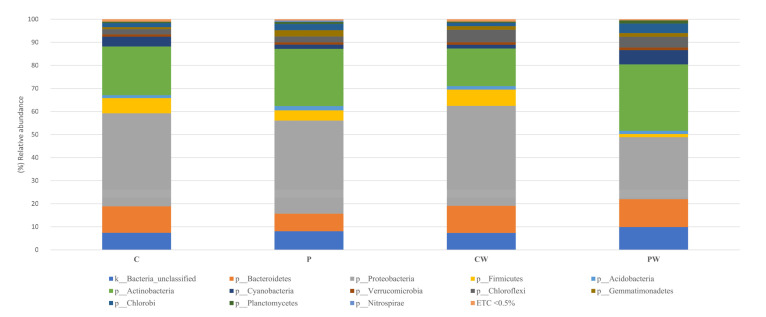
Relative abundance (%) of bacteria at the phylum in the skin mucus microbiota of fish feed with control diet without (C) and with wound (CW); fish feed with probiotic diet without (P) and with wound (PW). ETC: relative abundance <0.5% in average.

**Figure 3 animals-13-00193-f003:**
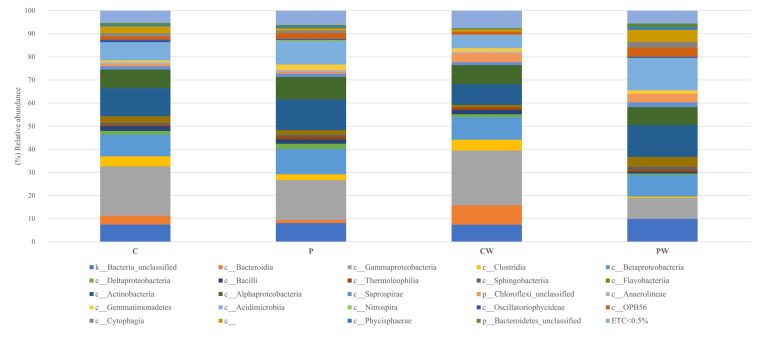
Relative abundance (%) of bacteria at class in the skin mucus microbiota of fish feed with control diet without (C) and with wound (CW); fish feed with probiotic diet without (P) and with wound (PW). ETC: relative abundance <0.5% in average; c__non taxonomy classify OTUs at this categorize.

**Figure 4 animals-13-00193-f004:**
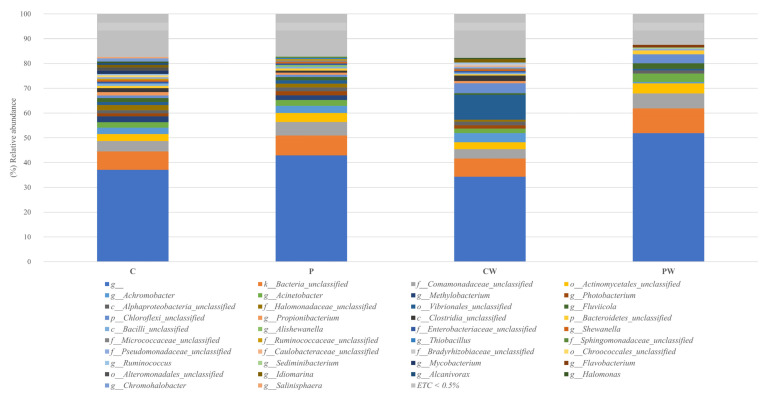
Relative abundance (%) of bacteria at genera level in the skin mucus microbiota of fish feed with control diet without (C) and with wound (CW); fish feed with probiotic diet without (P) and with wound (PW). ETC: relative abundance <0.5% in average; g__non taxonomy classify OTUs at this categorize.

**Figure 5 animals-13-00193-f005:**
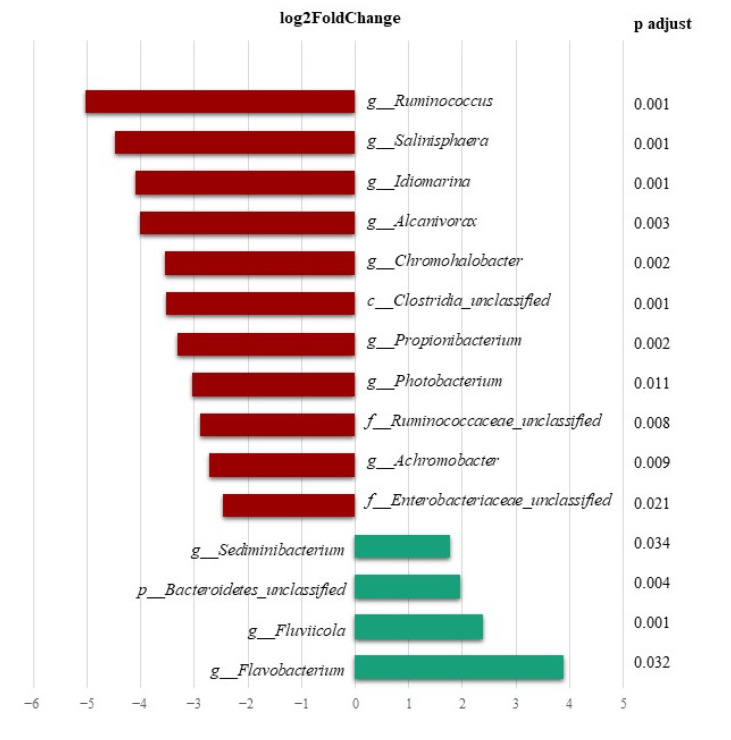
Significant differences of DESeq2 at the genus level with abundance relative > 0.5% (DESeq2, *p* < 0.05) in the mucus microbiota of the skin. Comparison between the wounded groups of fish that received the control diet (CW) and probiotic (PW).

**Figure 6 animals-13-00193-f006:**
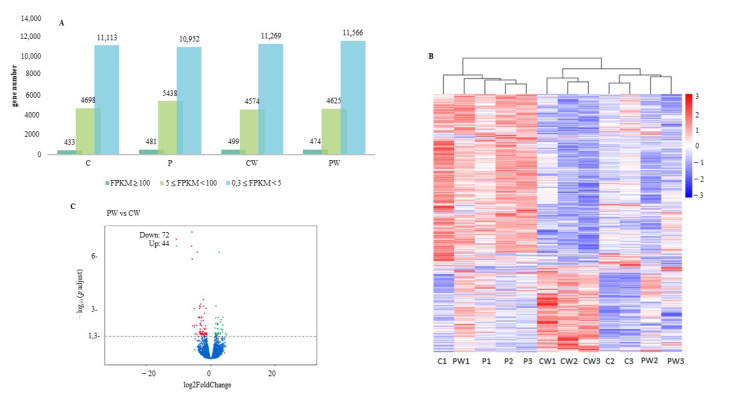
(**A**) Number of genes expressed during each stage with an average number of reads per kilobase per million reads (FPKM) of ≥ 0.3. (**B**) Heatmap by sample using the log10 (FPKM + 1) value. Red indicates genes with high levels of expression, while blue indicates low levels. The red to blue color scale represents the log10 (FPKM + 1) value from highest to lowest. (**C**) Volcano plots of differentially expressed genes (DEGs) between PW vs. CW comparison, expression data plotted with correct *p* value cutoff FDR < 0.05 and log_2_ fold change absolute value ≥ 1.3 Fish feed with control diet without (C) and with wound (CW). Fish feed with probiotic diet without (P) and with wound (PW). 1–3 represent biological replicates.

**Figure 7 animals-13-00193-f007:**
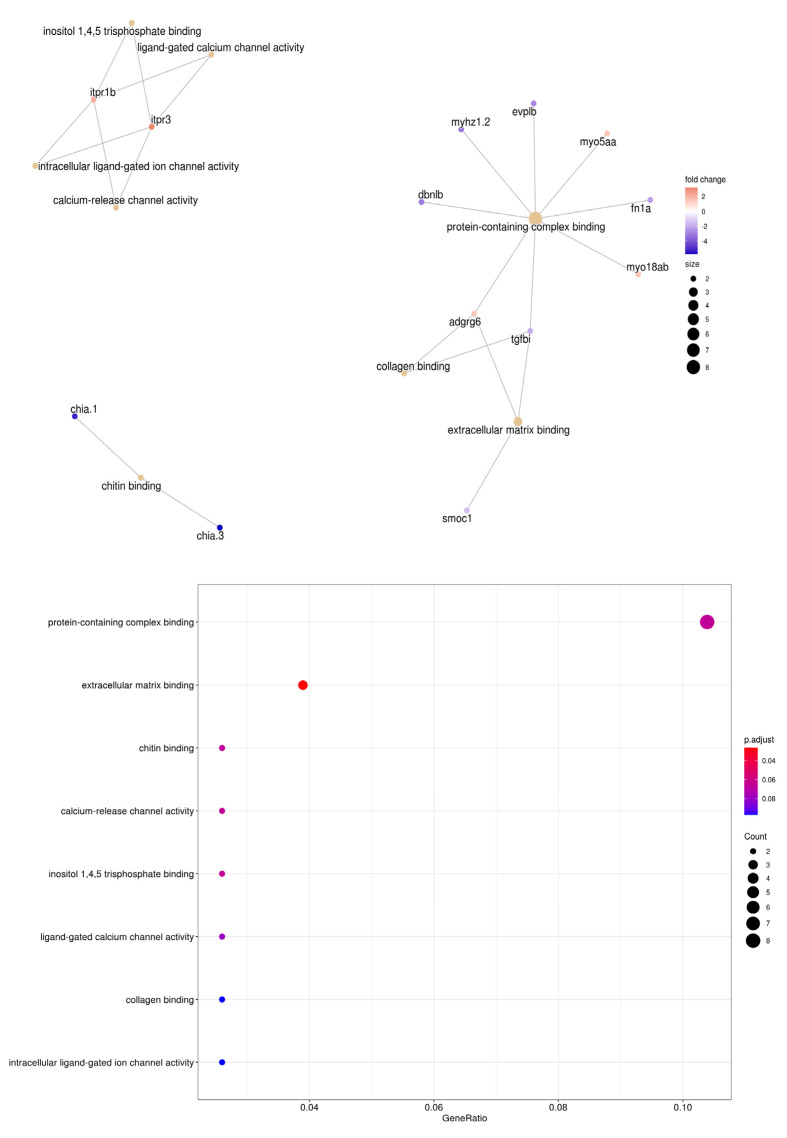
Significant functional enrichment network of the Biological process GO (adjusted *p* < 0.05), from the DEGs obtained in the comparison of the CW vs. PW skin. Brown nodes indicates functional categories. Red to blue nodes indicate the log_2_ fold change values.

**Figure 8 animals-13-00193-f008:**
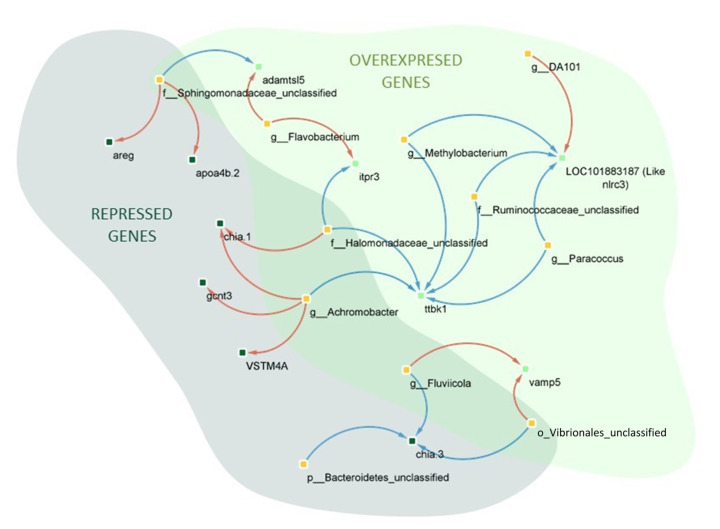
Correlation network gene (green nodes)—OTU (orange nodes) significant (Spearman, *p* < 0.05 and r < |0.9|, *p* < 0.05) based on DEG with the highest change value in the PW vs. CW comparison. Red lines indicate positive correlations while blue lines show negative correlations.

**Table 1 animals-13-00193-t001:** Alpha diversity indexes of the microbiota associated with the skin mucosa of *Sparus aurata*. Fish feed with control diet without (C) and with wound (CW) and fish feed with probiotic diet without (P) and with wound (PW).

	Chao1	Shannon	Simpson
C	743.69 ± 56.03	4.48 ± 0.28	0.97 ± 0.01
P	685.44 ± 73.15	3.87 ± 1.02	0.88 ± 0.18
CW	717.13 ± 93.15	4.02 ± 0.61	0.93 ± 0.07
PW	689.67 ± 109.17	3.87 ± 0.27	0.95 ± 0.01

The values represent the mean ± SD. Two-way ANOVA revealed not significant differences (*p* > 0.05).

**Table 2 animals-13-00193-t002:** Top 20 genes found to be significantly up- and down- regulated in PW compared to CW.

	*S.aurata* IDs	*D. renio* Ensembl IDs	logFC	Symbol Gene	Description
Inmune represor	LOC115567352	ENSDARG00000105739	4.535	LOC101883187	Like NLRC3, negative regulator of the innate immune response
Replication	LOC115579326	ENSDARG00000022280	4.847	*brd2a*	Bromodomain containing 2a
	LOC115569844	ENSDARG00000075560	2.548	*kmt2cb*	Lysine (K)-specific methyltransferase 2Cb
	LOC115575230	ENSDARG00000089292	2.361	*adgrl1a*	Adhesion G protein-coupled receptor L1a
Celular signal	adamtsl5	ENSDARG00000052118	3.74	*adamtsl5*	ADAMTS like 5
	itpr3	ENSDARG00000061741	3.227	*itpr3*	Inositol 1,4,5-trisphosphate receptor, type 3
Celular Migration	smpd3	ENSDARG00000098226	3.658	*smpd3*	Sphingomyelin phosphodiesterase 3
	LOC115582399	ENSDARG00000068262	3.237	*vamp5*	Vesicle-associated membrane protein 5
	LOC115581334	ENSDARG00000109620	3.061	*pxna*	Paxillin a
	ttbk1	ENSDARG00000056019	2.96	*ttbk1a*	Tau tubulin kinase 1a
Apoliprotein	LOC115578371	ENSDARG00000020866	−6.008	*apoa4b.2*	Apolipoprotein A-IV b, tandem duplicate 2
	LOC115579335	ENSDARG00000040295	−5.883	*apoeb*	Apolipoprotein Eb
	LOC115574970	ENSDARG00000073718	−4.828	*apol*	Apolipoprotein L
Quinases	LOC115583589	ENSDARG00000009612	−5.773	*chia.3*	Chitinase, acidic.3
	LOC115583106	ENSDARG00000100635	−5.216	*chia.1*	Chitinase, acidic.1
Cell proliferation	LOC115581553	ENSDARG00000076853	−4.973	*areg*	Amphiregulin
	meltf	ENSDARG00000075159	−5.649	*meltf*	Melanotransferrin
	LOC115590067	ENSDARG00000010154	−4.216	*vstm4a*	V-set and transmembrane domain containing 4a
Mucin biosynthesis	gcnt3	ENSDARG00000060471	−4.399	*gcnt3*	Glucosaminyl (N-acetyl) transferase 3, mucin type
Cell structure	LOC115570587	ENSDARG00000075441	−4.33	*vwa2*	Von Willebrand factor A domain containing 2

## Data Availability

The raw FASTQ files are currently available at the National Centre for Biotechnology Information (NCBI) BioProjects under the accession numbers PRJNA828434 and PRJNA862713.

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
