# Peer review of "SpPdp11 Administration in Diet Modified the Transcriptomic Response and Its Microbiota Associated in Mechanically Induced Wound Sparus aurata Skin"

_animals, 2023, doi:10.3390/ani13020193_

Round 1

Reviewer 1 Report

MS: Microbiota-tissue interaction in the skin healing of ulcered Sparus aurata after dietary administration of the probiotic SpPdp11

Manuscript ID: animals-2024574

Summary:

The aim of the paper was to evaluate if supplementing the diet with probiotic Shewanella putrefaciens (SpPdp11) modified the microbiota and transcriptome of skin wounds in Gilthead seabream. The authors showed that the wounded skin of fish supplemented with SpPdp11 (PW group) had a significantly lower abundance of Firmicutes, Actinobacteria and Proteobacteria, and a significantly higher abundance of Bacteroidetes. The same fish (PW) had significant up-regulation of transcripts associated with cell signalling, cell migration and the innate immune system, and a significant down-regulation of transcripts associated with inflammation and tissue repair, cell proliferation and transcripts associated with extracellular matrix. This is an interesting and relevant topic since skin lesions are frequently found in fish and may contribute to the decrease of overall fish welfare and to the decreased perception of quality by consumers in the context of aquaculture industry. And it is very interesting to evaluate how diet supplementation with probiotics may impact on fish skin lesions.

General concept comments:

Although the topic of the manuscript is relevant for the field and well-structured (introduction, methods, results, discussion and conclusions), unfortunately there are some weaknesses that make the outcomes of the manuscript less clear.

Introduction: It is difficult to find the focus of the manuscript since the introduction is lacking some organization. It begins with the detriment outcomes of fish skin ulcers, and only after that the roles of fish skin are addressed. There is almost no focus on fish skin microbiota and how it is linked to skin wounds, leaving out of the literature review recent findings on this topic in other fish. The objective of the study is not entirely clear; at the end of the introduction the authors stated that they were aiming the wound healing process, which is not in accordance with the overall work, since it consisted in the evaluation of the impact of probiotic SpPdp11 in the microbiota of wounded versus non-wounded skin, regardless the timing of the regeneration process.

Methods: My biggest concern is related to the methodology used. I have major concerns about the groups and how the fish were handled. For instance, why only half the fish of each tank were subjected to skin damage? Does that mean the other fish of the same tank was used as the non-damaged fish? If so, these fish were exposed to stressful conditions and as the authors stated in the introduction, there might be an effect of stress on the microbiota. How many replicate tanks are there per group?

Another major concern is related to the type of wound and the anatomical location of skin samples. Throughout the introduction, the authors demonstrated the problematic of fish skin ulcers (it is even highlighted in the title), but the skin samples analysed were mechanically induced wounds that only affected the epithelial layers of the skin. Usually, ulcers typically involve scale loss and may expose the underlying muscle. If the focus was skin ulcers, fish and skin samples should have been chosen based on already showing ulcers. It is important to make it clear what type of wound is the focus of the study. Also, what was the purpose of repeated skin damage? More importantly, skin samples from below the lateral line was not the best option. From Figure 1 there is the perception that skin samples were taken too close to the anus and there might be a contamination from the gut. Are there any samples of gut for microbiota analysis? Furthermore, it was not clear if the samples analysed were the mucus or the skin tissue. There was also an inconsistency about the number of replicates used per group (sometimes authors refer 6 fish, then 4 fish and at the end it is 3 fish).

Another major concern about the methodology of this work is that authors should have included samples of tank water for analysis of microbiota and quantitative analysis of 16S transcript.

One of the first questions raised when reading this manuscript was about the impact of SpPdp11 supplemented diet on the abundance of this bacteria on fish skin. If the bacteria is usually present in the skin of Gilthead seabream (it was even isolated from the skin of healthy fish to formulate the supplemented diet), is it affected by skin wounds? Is it affected by the probiotic supplemented diet? It would have been important to address these questions.

Results and discussion: There was a substantial lack of comparison between the groups. The authors only focused on the comparison between CW and PW groups, sometimes only referring to the results of PW. Moreover, there are no macroscopic nor microscopic images of the wounds. Taken all these weaknesses together, the conclusions were a bit ambitious, specifically when stating that skin lesions in PW group were in a more advanced stage of the wound healing. The results obtained cannot support this affirmation. The discussion is just an explanation of the genera and transcripts up or down-regulated. There is no connection between the microbiota and DEG results. There is no simple and direct information of what the meaning of the results is. Also, it was quite difficult to analyse the figures, specifically since the text size was too small to read it.

Conclusion: The outcomes of this work are substantially missing. Improvement of the discussion and solving the other methodological inaccuracies will assist to draw the conclusions.

Specific comments:

L27-28: There is at least two papers published about this topic.

L32: The observations that led to this suggestion must be included.

L34: Be specific about the diet and the genera.

L45-50: It is difficult to understand these sentences.

L58: What is the relevant ecological role of fish skin?

L63-64: This sentence breaks the connection between the former and the next sentences.

L92-93: The end of the sentence is out of place; difficult to understand the connection with the transcriptome analysis.

L117: How many replicate tanks are per group?

L119: What is the meaning of “rate” and “trial”?

L123: For how long the damage to the skin was performed? This information is obtained only in the legend of figure 1.

L126: It would be easier to understand the groups if this sentence appears in the first place.

L132: How were the samples divided? And why?

L140: I suggest consistency throughout the MS in the groups labels (sometimes it is ulcered and sometimes it is wounded). I believe wounded is more appropriate.

L144-145: What was the sample used for analysis? Why was the mucus “extracted” after sampling and not during sampling?

L166: I believe it is skin microbiota, not intestinal microbiota.

L176: Is it dietary treatment or per group?

L225: There is an inconsistency about the number of replicates per groups.

L236: Either choose the number of the additional file or the label of the supplementary file, to be easier to find it.

L265: This is methods.

L289: Both CW and PW groups?

L318: It would be better to have all the groups reflected in Table 2.

L352: It is impossible to read the figure.

L369: Again, the authors are focusing on the aspect of an ulcer lesion, but they analysed a mechanically induced wound.

L376: Citation 46 is not adequate for this argument.

L393-395: What about in gilthead seabream?

L482-495: What is the actually meaning of this positive and negative correlations? Can they be grouped in functional groups?

L502-504: Why the authors believe these changes are related to the intestinal microbiota? It is because they suspect a contamination from the gut microbiota?

Author Response

General concept comments:

Although the topic of the manuscript is relevant for the field and well-structured (introduction, methods, results, discussion and conclusions), unfortunately there are some weaknesses that make the outcomes of the manuscript less clear.

The authors are deeply grateful for the reviewer’s comments and suggestions. The lines referenced in our answer are the once from the revised version of the manuscript. Please see the attachment MS, the red letters correspond to the changes made for reviewer 1.  In addition, the authors are ready to take on any other suggestions from the reviewer.

Introduction: It is difficult to find the focus of the manuscript since the introduction is lacking some organization. It begins with the detriment outcomes of fish skin ulcers, and only after that the roles of fish skin are addressed. There is almost no focus on fish skin microbiota and how it is linked to skin wounds, leaving out of the literature review recent findings on this topic in other fish. The objective of the study is not entirely clear; at the end of the introduction the authors stated that they were aiming the wound healing process, which is not in accordance with the overall work, since it consisted in the evaluation of the impact of probiotic SpPdp11 in the microbiota of wounded versus non-wounded skin, regardless the timing of the regeneration process.

The authors would like to thank the reviewer for his suggestions on how to improve this manuscript. The introduction has been restructured as indicated and the objective has been modified (lines 46-100).

Methods: My biggest concern is related to the methodology used. I have major concerns about the groups and how the fish were handled.

The authors thank the reviewer for questions regarding the material and methods, as they have found a large error in the applied methodology. Animals were wounded only once, all details are written in the manuscript (line 112-148). We will also respond to the questions raised in this section.

For instance, why only half the fish of each tank were subjected to skin damage? Does that mean the other fish of the same tank was used as the non-damaged fish? If so, these fish were exposed to stressful conditions and as the authors stated in the introduction, there might be an effect of stress on the microbiota.

This methodology has been revised, the methodology that was actually carried out was as follows “The remaining fish (eight specimens from each diet, four from each aquarium) were anesthetized (20 mg L-1 of clove oil) and similar 8 mm diameter and 2 mm deep wounds were made in the middle of the left flank (below the lateral line) on the skin of all of them, with a circular biopsy punch (Integra™ Miltex™), taking care in avoiding any possible contamination with urinogenital and/or intestinal excretions. The animals remined in their respective aquaria receiving their corresponding diet for an extra seven days. The fish were fed the initially assigned diet, being the daily rate adjusted accordingly to the fish weight. After 37 days of total feed trial, the remaining fish (eight specimens from each diet and aquarium) were also sampled (CW and PW groups). Skin samples were obtained from the same body part of injured and uninjured fish, as shown in Figure 1.”

How many replicate tanks are there per group?

As it was indicated in the manuscript “Two tanks of 12 fish (two replicates) were fed each one of the two experimental diets at the previous indicated rate and twice a day”. That is, per diet there are two tanks per replicate, but per group, each replicate being the individuals.

Another major concern is related to the type of wound and the anatomical location of skin samples. Throughout the introduction, the authors demonstrated the problematic of fish skin ulcers (it is even highlighted in the title), but the skin samples analysed were mechanically induced wounds that only affected the epithelial layers of the skin. Usually, ulcers typically involve scale loss and may expose the underlying muscle. If the focus was skin ulcers, fish and skin samples should have been chosen based on already showing ulcers. It is important to make it clear what type of wound is the focus of the study.

The aim of this work was to study mechanically damaged skin. The injury was mechanically induced and affected epithelial layers of the skin but also the dermis. Of course, scale loss was also involved but the underlying muscle was never observed. Now these observations have been included in the materials and methods section. 

Also, what was the purpose of repeated skin damage? More importantly, skin samples from below the lateral line was not the best option. From Figure 1 there is the perception that skin samples were taken too close to the anus and there might be a contamination from the gut.

As mentioned above, the material and methods have been reviewed. In this experiment, the damage was not repeated and figure 1 shows how the sampling was performed. Also, special care was taken not to contaminate the sample (added line 131).

Are there any samples of gut for microbiota analysis?

Gut samples are not available as they were not part of the objective of this work. It would be very interesting for future work to see how gut microbiota correlates with skin transcriptomics when fish are supplemented with a probiotic diet.

Furthermore, it was not clear if the samples analysed were the mucus or the skin tissue.

The samples studied were from the microbiota analysis were mucus, “The skin samples, stored at -80 ° C were thawed gradually on ice and the mucus contents were extracted” (modified line 151). For RNA analyses were skin as indicated in the original manuscript, “Total RNA was extracted from gilthead seabream skin samples.” line 182.

There was also an inconsistency about the number of replicates used per group (sometimes authors refer 6 fish, then 4 fish and at the end it is 3 fish).

In line with the question, the number of replicates has been clarified in the manuscript. It was started from 4 replicates per group as mentioned in the original manuscript ("Skin samples from 4 fish per experimental group were randomly selected and each sample was split in two for further RNA and DNA analysis and stored at -80 °C until use"). For microbiota analysis it was stated "The skin samples (n=4 per group)". Now, section 2.4 was modified, the line “Only three of the samples passed the sequencing quality control, so an n=3 per group was kept for RNA seq analysis.” Line was added.

Another major concern about the methodology of this work is that authors should have included samples of tank water for analysis of microbiota and quantitative analysis of 16S transcript.

Furthermore, the microbiota associated with the skin is selective, and not all microorganisms that live in water bind to the mucus, as the characteristics of the mucus create their own ecosystem.

One of the first questions raised when reading this manuscript was about the impact of SpPdp11 supplemented diet on the abundance of this bacteria on fish skin. If the bacteria is usually present in the skin of Gilthead seabream (it was even isolated from the skin of healthy fish to formulate the supplemented diet), is it affected by skin wounds? Is it affected by the probiotic supplemented diet? It would have been important to address these questions.

Firstly, the genus Shewanella has been detected in the study in Sparus aurata skin in the different groups. However, significant differences were only obtained in the P vs PW comparison, showing a decrease in the genus Shewanella in damaged fish (table s1).

However, it should be noted that bioinformatic analysis using 16S rRNA amplicon does not allow reliable discrimination to species level, so we cannot confirm whether the SpPdp11 strain is present and/or there are changes in its abundance. abundance. Future shotgun metagenomic analyses may allow us to find out.  This is one of the weak points of the group, so tests are being carried out to detect and quantify the strain reliably, using other molecular techniques or identifying marker genes only present in it (Seoane et al., 2019).

Seoane, P., Tapia-Paniagua, S. T., Bautista, R., Alcaide, E., Esteve, C., Martínez-Manzanares, E., & Moriñigo, M. A. (2019). TarSynFlow, a workflow for bacterial genome comparisons that revealed genes putatively involved in the probiotic character of Shewanella putrefaciens strain Pdp11. PeerJ, 7, e6526.

Results and discussion: There was a substantial lack of comparison between the groups. The authors only focused on the comparison between CW and PW groups, sometimes only referring to the results of PW. Moreover, there are no macroscopic nor microscopic images of the wounds. Taken all these weaknesses together, the conclusions were a bit ambitious, specifically when stating that skin lesions in PW group were in a more advanced stage of the wound healing. The results obtained cannot support this affirmation.

This work is based on the conclusions obtained by Chen et al 2020 (macroscopic images of the wound were provided), obtaining results that complement the hypothesis made by this author. This point has been clarified in the introduction (lines 88-97).

In addition, in the skin-associated microbiota study, the group that produced the most significant changes was the PW, with a different microbial profile to the other study groups. As the aim of the work was to study the differences in wounded skin when the diet is supplementing with a probiotic, if we were to describe all the results, the work would be too long and without a clear focus. Therefore, we decided to take the CW and PW groups to see in more detail how, from the same wound, the change in diet produced a transcriptional modification, which genes were modified and the microbiota associated with it.

The discussion is just an explanation of the genera and transcripts up or down-regulated. There is no connection between the microbiota and DEG results. There is no simple and direct information of what the meaning of the results is.

We thank the reviewer for this comment to improve the discussion of this manuscript. The discussion section has been modified in red in the new attached manuscript.

Also, it was quite difficult to analyse the figures, specifically since the text size was too small to read it.

The figures have been pasted into the manuscript and this may have diminished their quality. The authors have increased the size of the images to make them more readable in the text. Likewise, all figures should be attached with the highest possible quality so that the journal can include them at the end of the correction. To improve the compression of figure 7, it has been modified, and section A and C has been moved to supplementary material (Figure S6A and S6B respectively).

Conclusion: The outcomes of this work are substantially missing. Improvement of the discussion and solving the other methodological inaccuracies will assist to draw the conclusions.

Both sections have been modified in the new manuscript, all changes are shown in red.

Specific comments:

L27-28: There is at least two papers published about this topic.

Sentence was modified “ Knowing that the use of probiotics can improve fish skin health, SpPdp11 dietary administration has demonstrated beneficial effects for farmed fish, so its potential on the skin needs to be studied more deeply.”

L32: The observations that led to this suggestion must be included.

Sentence has been modified: “The wounded specimens which received the diet with SpPdp11 showed a decrease in the abundance of Enterobacteriaceae, Photobacterium and Achromobacter related to bacterial biofilm formation, as well as the overexpression of genes involved in signaling mechanisms (itpr3), cell migration and differentiation (panxa, ttbk1a, smpd3, vamp5); and repression of genes related to cell proliferation (vstm4a, areg), consistent with a more efficient skin healing processes than that observed in the wounded control group.”

L34: Be specific about the diet and the genera.

The sentence has been completed: In addition, among the groups of damaged skin with different diets, Achromobacter, f_Ruminococcaceae, p_Bacteroidetes, Fluviicola and Flavobacterium genera with significant differences showed positive correlations with genes related to cell migration and negative correlations with cytokines and cell proliferation and may be the target of future studies.

L45-50: It is difficult to understand these sentences.

Sentence has been removed

L58: What is the relevant ecological role of fish skin?

Reverter et al. published “External fish mucus surfaces also play important roles in social relationships between conspecifics (fish shoaling, spawning synchronisation, suitable habitat finding, or alarm signals) and in interspecific interactions such as prey-predator relationships, parasite host interactions, and symbiosis”

This line (now line 73-76) has been modified in relation to the reference used: “For these reasons, there is an increased interest in the fish skin health in recent years as it provides mechanical protection against physical, chemical and biological damage and mucus plays a relevant ecological role[19,20]”.

Reverter, M., Tapissier-Bontemps, N., Lecchini, D., Banaigs, B., & Sasal, P. (2018). Biological and ecological roles of external fish mucus: a review. Fishes3(4), 41. (Number 20 in references of MS)

L63-64: This sentence breaks the connection between the former and the next sentences.

Thanks for the appreciation, the phrase has been included in the new context, making more sense.

L92-93: The end of the sentence is out of place; difficult to understand the connection with the transcriptome analysis.

The lines 94-97 has been added to improve the connection with the target

L117: How many replicate tanks are per group?

There were two replicates (two tanks) per group. To improve the comprehensibility of this sentence, the lines 115- 119 have been modified.

L119: What is the meaning of “rate” and “trial”?

Rate is defined as a measure, quantity or frequency. The word trial was replaced by " assay " for a better understanding.

L123: For how long the damage to the skin was performed? This information is obtained only in the legend of figure 1.

The authors are grateful for the reviewer's interest in the material and methods, which has enabled them to modify their error. The damage was done only once as shown on line 126 -133.

L126: It would be easier to understand the groups if this sentence appears in the first place.

As the reviewer indicates, the sentence has been moved

L132: How were the samples divided? And why?

L144-145: What was the sample used for analysis? Why was the mucus “extracted” after sampling and not during sampling?

The samples were split in half so that they could be frozen independently. In this way, RNA and DNA extraction could be carried out independently and more carefully.

We wanted to obtain results from the skin mucus-associated microbiota, which was as close as possible to the tissue used for RNA seq. That is why we decided to take one sample and split it in two. As one part was to be used for RNA seq, the sample was taken as quickly as possible, and cryopreserved so as not to alter the quantity and quality of the RNA. Later, at the facilities of the University of Malaga, the samples to microbiota analysis were manipulated to extract the mucus from the skin.

L140: I suggest consistency throughout the MS in the groups labels (sometimes it is ulcered and sometimes it is wounded). I believe wounded is more appropriate.

The word “ulcered” has been changed in the manuscript to “wounded”.

L166: I believe it is skin microbiota, not intestinal microbiota.

The authors apologise for the error

L176: Is it dietary treatment or per group?

The authors apologise for the error, it is per group

L225: There is an inconsistency about the number of replicates per groups.

This answer has been answered before. The authors have clarified this point in the new version of the manuscript.

L236: Either choose the number of the additional file or the label of the supplementary file, to be easier to find it.

According to the reviewer, the nomenclature "additional file" has been removed.

L265: This is methods.

According to the reviewer, the sentence "DESeq2 analysis was applied to assess significant differences (adjustment p < 0.05) between the abundance of microbial taxa" has been deleted and amended to "No differences were detected between fish (DESeq2, p < 0.05)".

L289: Both CW and PW groups?

The authors do not fully understand this question and ask the reviewer if it is possible to clarify it.

L318: It would be better to have all the groups reflected in Table 2.

As mentioned above, based on the objective of the work and the reader's understanding, the PW and CW comparison was shown. Including all comparisons in this table would not add a line to the manuscript.

Line 316 and 318 was added

L352: It is impossible to read the figure.

To improve the compression of figure 7, it has been modified, and section A and C has been moved to supplementary material (Figure S6A and S6B respectively).

L369: Again, the authors are focusing on the aspect of an ulcer lesion, but they analysed a mechanically induced wound.

The sentence “It possible that the realization of a mechanical wound does not imply changes in microbial diversity, that does happen when ulcers are produced by biological agents and its interactions with the microbiota” was added to improve the discussion (line 396 -398)

L376: Citation 46 is not adequate for this argument.

The reference has been deleted

L393-395: What about in gilthead seabream?

The authors have found no evidence. Line 419-422 was added to MS: “This fact has not studied in Sparus aurata but in the future, some studies could be focused to corroborate the relation between the presence of those taxa and epithelial cell proliferation”

L482-495: What is the actually meaning of this positive and negative correlations? Can they be grouped in functional groups?

A positive correlation shows that the presence or increase of a certain micro-organism is significantly related to the presence or increase of a gene. On the other hand, a negative correlation shows that the absence or decrease of a micro-organism is significantly related to the presence or increase of a gene.

 Functional groups can only be observed with certainty when we observe the functional enrichment of the genes studied.

L502-504: Why the authors believe these changes are related to the intestinal microbiota? It is because they suspect a contamination from the gut microbiota?

The authors do not suspect contamination of the samples at any time, as special care was taken.

We must not forget that the probiotic is taken at the level of the digestive tract, so indirectly the intestinal microbiota is affecting the skin of the individuals who took this diet. The study by Naya-Catala (2021) also reported a systemic effect of the intestinal microbiota on other organs, in this case the liver and kidney.

Reviewer 2 Report

This MS is about effect of probiotic SpPdp11 on transcriptome and microbiota response of wound skin of Sparus aurata.

Study design and results are interesting and understandable although there is lacking points with low number of replications.

Conclusion was that probiotic SpPdp11 supplemented to gut can involve in skin wound healing. 

I recommand this MS to be a publication after few changes.

Conclusion: 

About the concept of gut-skin axis, it would be more nicer if you compare transcriptome and microbiota between gut and skin in this model.

Please mention your future perspectives and research plan on this points.

Author Response

The authors are very grateful for the review. The reviewer's suggestion is very interesting and will be taken into account for future work.

In future experiments, gut and skin samples would be taken in addition to the skin transcriptomic samples. We would like to correlate whether the presence or absence of certain microorganisms in the gut, when modulated by the probiotic, directly influences skin transcriptomics.

We also plan to do future experiments in which we can take metabolomic samples from the gut and see directly which metabolisms may be involved in skin regeneration.

Round 2

Reviewer 1 Report

Comments to Revised MS “SpPdp11 administration in diet modified the transcriptomic response and its microbiota associated in mechanically induced wound Sparus aurata skin.”

Summary

Overall, the authors addressed the comments to the submitted MS and were able to seriously revise and improve the quality of the MS, specifically the Introduction and Materials and Methods. I still have some comments that could help the authors to improve a bit more the MS, specifically at the level of the Discussion and Conclusion.

1 – Groups and replicates:

I still do not understand how many replicate tanks are per group. Perhaps, it is just a manner of rephrasing the text. But from the text and the authors reply, it seems that each group only has one tank, therefore no replicate tanks per diet/group.

2 – Skin wound and sampling:

It is important to explain in the methods that skin and mucus samples were taken from a different anatomical site than the injury/wound. I only understood that from figure 1 subtitle. In addition, I suggest using different colours for the site of injury/wound and for the site of skin sampling (figure 1).

3 – Results/statistical analysis:

I maintain my previous observation that there is a substantial lack of comparison between the groups. The authors stated “the group that produced the most significant changes was the PW, with a different microbial profile to the other study groups” and this should be more important than what they present in L316. Even if the aim is to investigate the effect of the probiotic diet on the microbiota profile in the wounded fish, the authors cannot ignore the other groups (C and P), specifically at the level of the statistics. That is why I do not agree t-student test is suitable. Instead, authors should use two-way ANOVA, as they have two factors in this experiment (diet and wound) and they want to investigate if there is an interaction between diet and wound. The statistical analysis loses power if the other groups are taken away from the statistical analysis.

4 – Discussion and conclusion:

Discussion is too long and lacking a more focused interpretation of the results. The conclusion was not modified and thus still misses the outcomes of this work. It will be easier to draw the conclusions after improving the discussion. Although it is important to give a basic information about transcripts and microbiota, it is easy to get lost while reading the discussion with so much information. Also, the authors should have in mind that they are analysing microbiota and skin from a wounded fish but not from wounded skin samples, so it might be too pushy when stating that PW skin is at a more advanced stage of the wound healing than the CW (L492-L497), specially when they did not have macroscopic observations of the wound.

The conclusion is not a clear message about the outcomes of this study. It does not answer to the aim of the study stated in the introduction. What was the effect of the probiotic diet on the microbiota of wounded skin? Was there an effect of wound in the microbiota of wounded skin in the absence of the probiotic diet? If so, was the same effect observed in the PW group? If not, how were they different? A direct and simple answer to theses questions would allow an understanding of the outcomes of this study.

Minor points:

L45: “farming” is repeated and consider using either “important” or “significant”.

L74-76: Something is missing in the sentence.

L87: There is a repetition of “diet”.

L117-120: A bit confusing using “fundamental and basic feed”. Simplify it.

L121-125: Repetition of sentences.

L134: Consider stating “Seven days after the wounding” or similar, instead of 37 days to avoid confusion.

L143: Prefer the use of objective information as the number of fish instead of expressions as “half of the fish”.~

L152: Consider adding mucus sample.

L158: Consider adding mucus sample.

L380-388: I suggest opening the discussion with the main findings of this study, not others.

L393-395: I do not think 47 is an appropriate reference here (no wounds in that reference). Also, there is a substantial difference between this study and those cited in references 1 and 21. In those, samples were collected from the wound and in this study, samples were taken in a different place. The subsequent argument needs to be revised in agreement. And this should be taken in mind throughout the discussion.

L556-561: I do not think this argument is relevant to the discussion.

Author Response

Summary

Overall, the authors addressed the comments to the submitted MS and were able to seriously revise and improve the quality of the MS, specifically the Introduction and Materials and Methods. I still have some comments that could help the authors to improve a bit more the MS, specifically at the level of the Discussion and Conclusion.

The authors are again grateful for the reviewer's comments. We will respond to the questions asked in order to improve the MS.

1 – Groups and replicates:

I still do not understand how many replicate tanks are per group. Perhaps, it is just a manner of rephrasing the text. But from the text and the author's reply, it seems that each group only has one tank, therefore no replicate tanks per diet/group.
Four tanks/aquaria with the same number of fish (12 per tank) were used as a starting point. Two of the tanks were fed the control diet, while the other two were fed the probiotic diet. 8 fish fed the control diet (4 from each tank) and 8 fish fed the probiotic diet (4 from each tank) were sampled prior to damage, obtaining groups C and P. On the other hand, after 7 days of damage, 8 fish fed the probiotic diet (PW group) and 8 fish fed the control diet (CW group) (4 from each tank) were sampled. As the reviewer can see, there were fish left over, but they were not used, it was to ensure the randomness of the experiment.

The tanks diagram can be found at the end of the reference section in the attached manuscript; it can be added to the supplementary material if you consider it appropriate.

Paragraph 2.2 has been reviewed and revised in order to clarify the methodology used.

2 – Skin wound and sampling:

It is important to explain in the methods that skin and mucus samples were taken from a different anatomical site than the injury/wound. I only understood that from figure 1 subtitle. In addition, I suggest using different colours for the site of injury/wound and for the site of skin sampling (figure 1).

According to the reviewer's comment, line 124-125 was added. The color for figure 1 was also modified

3 – Results/statistical analysis:

I maintain my previous observation that there is a substantial lack of comparison between the groups. The authors stated “the group that produced the most significant changes was the PW, with a different microbial profile to the other study groups” and this should be more important than what they present in L316.

According to the reviewer's comment, line 320-324 was modified.

Even if the aim is to investigate the effect of the probiotic diet on the microbiota profile in the wounded fish, the authors cannot ignore the other groups (C and P), specifically at the level of the statistics. That is why I do not agree t-student test is suitable. Instead, authors should use two-way ANOVA, as they have two factors in this experiment (diet and wound) and they want to investigate if there is an interaction between diet and wound. The statistical analysis loses power if the other groups are taken away from the statistical analysis.

The t-test has been modified by a two-way ANOVA to employing probiotic and wounding as 2 factors. T-test was only used in the alpha-diversity analysis. As in the previously obtained results, no significant differences were observed in any of the cases.

The methodology (line 177) and results (line 237) have been modified in the manuscript and the ANOVA results have been added in Table S1.

4 – Discussion and conclusion:

Discussion is too long and lacking a more focused interpretation of the results. The conclusion was not modified and thus still misses the outcomes of this work. It will be easier to draw the conclusions after improving the discussion. Although it is important to give a basic information about transcripts and microbiota, it is easy to get lost while reading the discussion with so much information. Also, the authors should have in mind that they are analysing microbiota and skin from a wounded fish but not from wounded skin samples, so it might be too pushy when stating that PW skin is at a more advanced stage of the wound healing than the CW (L492-L497), specially when they did not have macroscopic observations of the wound.

The conclusion is not a clear message about the outcomes of this study. It does not answer to the aim of the study stated in the introduction. What was the effect of the probiotic diet on the microbiota of wounded skin? Was there an effect of wound in the microbiota of wounded skin in the absence of the probiotic diet? If so, was the same effect observed in the PW group? If not, how were they different? A direct and simple answer to theses questions would allow an understanding of the outcomes of this study.

Taking into account all the reviewer's comments, the discussion and conclusion have been further revised and modified. We hope that it satisfies the requirements.

Minor points:

L45: “farming” is repeated and consider using either “important” or “significant”.

The authors apologise for the error, the sentence has been modified: “The quick development of aquaculture due to the high demand for fish had led to intensive farming which conditions have a significant impact on fish's health”

L74-76: Something is missing in the sentence.

The sentence was modified: “For these reasons, there is an increased interest in the fish skin health in recent years as it provides mechanical protection against physical, chemical and biological damage. In this sense, mucus also plays a relevant ecological role as fish shoaling or alarm signals and in interspecific interactions such as prey-predator relationships, parasite-host interactions, and symbiosis [19,20]”.

L87: There is a repetition of “diet”.

Thanks to the reviewer for the detail.

L117-120: A bit confusing using “fundamental and basic feed”. Simplify it.
The sentence was modified: “The basal feed was the commercial diet, that was supplemented with equal volumes of phosphate saline buffer (PBS) or bacterial suspension (109 cfu SpPdp11 g-1) to create the control and probiotic diets respectively.

L121-125: Repetition of sentences.

The sentence was modified “The basal feed was the commercial diet, that was supplemented with equal volumes of phosphate saline buffer (PBS) or bacterial suspension to create the control (C, PBS) and probiotic (P, 109 cfu SpPdp11 g-1) diets respectively.”

L134: Consider stating “Seven days after the wounding” or similar, instead of 37 days to avoid confusion.
The sentence was modified: “Seven days after the wounding, the remaining fish…”

L143: Prefer the use of objective information as the number of fish instead of expressions as “half of the fish”.

The number of fish was clarified.

L152: Consider adding mucus sample.
According to the reviewer the change was done.

L158: Consider adding mucus sample.
According to the reviewer the change was done.

L380-388: I suggest opening the discussion with the main findings of this study, not others.

Based on the comment, the first paragraph of the discussion was deleted.

L393-395: I do not think 47 is an appropriate reference here (no wounds in that reference). Also, there is a substantial difference between this study and those cited in references 1 and 21. In those, samples were collected from the wound and in this study, samples were taken in a different place. The subsequent argument needs to be revised in agreement. And this should be taken in mind throughout the discussion.

The paragraph was modified to “Analysis of the skin mucus-associated microbiota of the different groups of fish showed that dietary administration of SpPdp11 and mechanical wounding did not induce significant changes in alpha diversity (Two-Way ANOVA, p < 0.05) regardless of the treatment analyzed. In contrast with these results where the mucus skin was not directly of the wound, others studies the diversity of the microbiota associated with the fish skin wounded by microbial infection was increased [1,21]. It possible that the realization of a mechanical wound does not imply changes in microbial diversity indirectly, that does happen when ulcers are produced by biological agents and its interactions with the microbiota.”

L556-561: I do not think this argument is relevant to the discussion.

The argument was removed.
